# The Small-Molecule Wnt Inhibitor ICG-001 Efficiently Inhibits Colorectal Cancer Stemness and Metastasis by Suppressing MEIS1 Expression

**DOI:** 10.3390/ijms222413413

**Published:** 2021-12-14

**Authors:** Jang-Hyun Choi, Tae-Young Jang, So-El Jeon, Jee-Heun Kim, Choong-Jae Lee, Hyeon-Ji Yun, Ji-Youn Jung, So-Yeon Park, Jeong-Seok Nam

**Affiliations:** 1School of Life Sciences, Gwangju Institute of Science and Technology, Gwangju 61005, Korea; wkdgus1216@gist.ac.kr (J.-H.C.); gistjty1300@gm.gist.ac.kr (T.-Y.J.); thdpf8939@gm.gist.ac.kr (S.-E.J.); jeeheun.kim@gist.ac.kr (J.-H.K.); cjlee7847@gist.ac.kr (C.-J.L.); hyeon1540@gm.gist.ac.kr (H.-J.Y.); 2Department of Companion and Laboratory Animal Science, Kongju National University, Gongju 32439, Korea; wangza@kongju.ac.kr; 3Cell Logistics Research Center, Gwangju Institute of Science and Technology, Gwangju 61005, Korea

**Keywords:** colorectal cancer, cancer stem-like cells, metastasis, Wnt, ICG-001, MEIS1

## Abstract

Recurrence and metastasis remain major obstacles in colorectal cancer (CRC) treatment. Recent studies suggest that a small subpopulation of cells with a self-renewal ability, called cancer stem-like cells (CSCs), promotes recurrence and metastasis in CRC. Unfortunately, no CSC inhibitor has been demonstrated to be more effective than existing chemotherapeutic drugs, resulting in a significant unmet need for effective CRC therapies. In this study, transcriptomic profiling of metastatic tumors from CRC patients revealed significant upregulation in the Wnt pathway and stemness genes. Thus, we examined the therapeutic effect of the small-molecule Wnt inhibitor ICG-001 on cancer stemness and metastasis. The ICG-001 treatment efficiently attenuated self-renewal activity and metastatic potential. Mechanistically, myeloid ecotropic viral insertion site 1 (MEIS1) was identified as a target gene of ICG-001 that is transcriptionally regulated by Wnt signaling. A series of functional analyses revealed that MEIS1 enhanced the CSC behavior and metastatic potential of the CRC cells. Collectively, our findings suggest that ICG-001 efficiently inhibits CRC stemness and metastasis by suppressing MEIS1 expression. These results provide a basis for the further clinical investigation of ICG-001 as a targeted therapy for CSCs, opening a new avenue for the development of novel Wnt inhibitors for the treatment of CRC metastasis.

## 1. Introduction

Colorectal cancer (CRC) is one of the most common malignant neoplasms and a major cause of cancer-related death worldwide [1]. Although advances have been made in the progression of chemotherapy and targeted therapy, metastasis remains a major obstacle in CRC treatment [2]. Cancer stem-like cells (CSCs) can self-renew, differentiate, and repopulate the entire heterogeneous population of cancer cells, thus accounting for cancer recurrence and metastasis [3]. Therefore, the development of CSC-targeting drugs could be a breakthrough in the treatment of CRC metastasis. 

CSC-related signaling pathways, such as the Hedgehog, Notch, and Wnt signaling pathways, have attracted great interest as therapeutic targets for CSCs [4]. Our previous studies demonstrated that Wnt signaling is critical for the maintenance of CSCs in breast and liver cancer [5,6,7]. In CRC, aberrant activation of Wnt signaling is considered an initiating step in cancer development and is associated with a poor prognosis [8,9]. Moreover, among the primary CRC cells isolated from patient tumors, the subpopulation with higher Wnt transcriptional activity exhibited CSC properties and had higher tumor-initiating potential when inoculated into mice [10]. These results suggest the potential importance of Wnt signaling in CRC stemness. However, no Wnt inhibitor has been approved for clinical use in CRC treatment.

ICG-001 was first identified in a screen of small molecules that inhibited Wnt transcriptional activity in CRC cell lines [11]. Recent studies have provided convincing evidence that ICG-001 treatment could have benefits for the suppression of tumor growth [12,13,14]. However, further preclinical study is required to ensure its effectiveness in treating CRC metastasis. Moreover, the exact molecular mechanism underlying the therapeutic effects of Wnt inhibition by ICG-001 remains elusive.

In this study, by exploring the transcriptomic profiles of tumor tissues from CRC patients, we confirmed the aberrant activation of Wnt signaling in metastatic tumors. Thus, we examined the therapeutic potential and related mechanism of ICG-001 in suppressing CRC metastasis and CSC properties by conducting multiple functional analyses in vitro and in vivo. Mechanistically, we identified MEIS1 as a novel target gene of ICG-001 that promotes CSC properties and the metastasis of CRC. Our findings suggest that ICG-001 is a potentially useful small-molecule therapeutic for targeting CSCs and may provide a basis for further clinical evaluation in the treatment of CRC metastasis.

## 2. Results

### 2.1. Increased Stemness and Wnt Activation Are Associated with CRC Metastasis

Wnt/β-catenin signaling has shown promising potential as a treatment target in multiple types of cancer, including CRC [8]. To investigate the relevance of Wnt signaling to CRC stemness, we enriched the CSC population by a sphere culture [15] using human CRC cell lines HCT116 and HT29 (Figure 1A). The RT-qPCR revealed significantly increased expression of stemness-related genes [16,17,18] in spheres compared with bulk cells (Figure 1A), confirming CSC enrichment by the sphere culture, as determined in our previous report [9]. Notably, in these CSC-enriched spheres, we observed global increases in the expression levels of Wnt-related genes (Figure 1A). Next, to confirm this phenomenon, we performed an immunofluorescence assay to visualize Lymphoid Enhancer Binding Factor 1 (LEF1), a transcription factor (TF) involved primarily in the activation of Wnt signaling [19], and OCT4, a stemness TF regulated by Wnt signaling [20]. Consistent with the above results, immunofluorescence confirmed the concomitant elevation of LEF1 and OCT4 expression in spheres compared with monolayer-cultured bulk cells (Figure 1B).

Next, to understand the phenotypic alterations occurring in CRC metastasis, we performed a gene set enrichment analysis (GSEA), comparing the transcriptomic footprint of metastatic CRC against the existing gene sets in MSigDB. First, we obtained the differentially expressed genes (DEGs) in metastasis by comparing the primary tumors of patients with metastatic (Dukes’ stage D) CRC and patients with early-stage nonmetastatic (Dukes’ stage A) CRC and conducted a GSEA with these DEGs (Figure 1C). Several oncogenic gene sets were identified and LEF1 target genes were one of the most significantly enriched gene sets associated with CRC metastasis (Figure 1C). Wnt signature genes showed global trends of increases in metastatic CRC compared with early-stage nonmetastatic CRC (Figure 1D, left). In addition, stemness signature genes were enriched in metastatic CRC compared with early-stage nonmetastatic CRC (Figure 1D, right). These results suggest that the gene alterations in metastatic CRC mimic the transcriptomic footprint of stemness and Wnt activation.

### 2.2. Targeting Wnt Signaling with ICG-001 Efficiently Attenuates CRC Stemness and Metastasis

Given the potential importance of aberrant Wnt activation in CRC metastasis, we examined the therapeutic efficacy of ICG-001, using the malignant human CRC cell line HCT116, which exhibits a high tumorigenic potential in vivo [21]. Since ICG-001 inhibits the growth of HCT116 cells with an IC_50_ of 5.57 μM (Figure 2A), we treated cells with ICG-001 at approximately half of the IC_50_, i.e., 2.5 μM. RT-qPCR revealed that ICG-001 treatment resulted in global reductions in stemness-related gene expression levels (Figure 2B). Next, we conducted a serial sphere formation assay to examine the self-renewal activity of CSCs *in vitro*. During primary sphere formation, ICG-001 treatment resulted in a dose-dependent reduction in sphere formation (Figure 2C). Then, we isolated single cells from spheres, replated them, and cultured them under the sphere culture conditions to generate secondary spheres without additional ICG-001 treatment. ICG-001 treatment significantly and irreversibly reduced the formation of secondary spheres (Figure 2C), suggesting its potent and irreversible inhibitory effect on the self-renewal activity of CSCs.

Next, we examined the in vivo therapeutic efficacy of ICG-001, using multiple HCT116 xenograft models. First, we performed a limiting dilution assay (LDA) to determine the self-renewal ability of CSCs based on their tumor-initiating potential in vivo [22]. HCT116 tumor-bearing mice were treated daily with ICG-001 (100 mg/kg) for 6 weeks. Then, single tumor cells were isolated from primary tumors and reinjected into mice without any additional ICG-001 treatment. The ICG-001 treatment significantly decreased the incidence of tumor formation in vivo (Figure 2D), suggesting an irreversible reduction in the self-renewal ability after ICG-001 treatment. Next, to examine the therapeutic effect of ICG-001 on CRC metastasis, we monitored the extent of liver metastasis in a mouse model established by a splenic injection of HCT116-luc cells. The ICG-001 treatment significantly inhibited metastatic outgrowth to the liver (Figure 2E). In addition, the tumor burden in the liver was efficiently reduced by the ICG-001 treatment (Figure 2F). Moreover, we analyzed the gene expression levels in metastatic tumors from mouse livers and confirmed a global reduction in the expression levels of metastasis- and CSC-related genes following the ICG-001 treatment (Figure 2G). Collectively, these results suggest that ICG-001 exerts potent anti-CSC activity and efficiently inhibits CRC metastasis.

### 2.3. MEIS1 Is a Potential Target Gene of ICG-001, Which Is Associated with CRC Stemness and Clinical Malignancy

To identify potential target genes of ICG-001 associated with stemness and CRC metastasis, we explored the transcriptomic data of CRC patients. Among the stemness signature genes (*n* = 261), we searched for a subset of genes significantly associated with malignant CRC using the R2 platform (Figure 3A). By this method, we identified 21 genes upregulated in metastatic (Dukes’ stage D) tumors compared with early-stage nonmetastatic (Dukes’ stage A) primary tumors (*p* < 0.005) and 77 genes upregulated in recurrent compared with nonrecurrent tumors (*p* < 0.005). In parallel, to identify potential Wnt target genes, we explored the genes correlated with LEF1 and β-catenin (CTNNB1), the TFs of Wnt signaling. We identified 103 genes positively correlated with LEF1 expression (*p* < 0.00001) and 65 genes positively correlated with β-catenin expression (*p* < 0.005). By overlapping these four gene sets, we identified six common genes and considered them candidate target genes for ICG-001 (Figure 3A). For validation, we analyzed the transcriptional changes in these candidate genes after the ICG-001 treatment and found that myeloid ecotropic viral insertion site 1 (MEIS1) showed the most significant reduction upon ICG-001 treatment in HCT116 cells (Figure 3B). Moreover, MEIS1 expression was significantly decreased by LEF1 and β-catenin knockdown (Figure 3C), supporting the positive correlation between MEIS1 and LEF1/β-catenin in tumors of CRC patients (Figure 3D). Moreover, the MEIS1 expression level in primary tumors was significantly associated with the degree of CRC progression, being elevated in advanced-stage CRC tissues compared with early-stage CRC tissues (Figure 3E). Furthermore, increased MEIS1 expression was associated with poorer recurrence-free survival in CRC patients (Figure 3F). In parallel, Western blot analyses have confirmed the elevation of MEIS1 expression in human CRC cell lines compared with a normal colon cell line (CCD-18Co, Figure 3G). Collectively, our findings suggest MEIS1 as a potential target gene of ICG-001 treatment associated with cancer stemness and CRC malignancy.

### 2.4. MEIS1 Overexpression Enhances the Self-Renewal Capacity of CSCs and Metastasis of CRC

To confirm the potential relevance of MEIS1 to cancer stemness, we examined the protein level of MEIS1 in CSC-enriched spheres. The immunofluorescence assay revealed elevation of the MEIS1 protein level in CSC-enriched spheres compared with monolayer-cultured bulk cells (Figure 4A). Next, to investigate the functional importance of MEIS1 in CRC stemness, we generated MEIS1-overexpressing (OE) HCT116 cells (Figure 4B). The RT-qPCR analyses confirmed the global increases in stemness-related gene expression levels upon MEIS1 overexpression (Figure 4C). Consistent with this result, in the sphere formation assay, MEIS1 overexpression increased the number of spheres formed (Figure 4D), whereas MEIS1 knockdown decreased the number of spheres formed (Appendix A), indicating that MEIS1 enhances the self-renewal activity of CSCs *in vitro*. To validate the observance of this phenomenon, we compared the self-renewal ability between wild-type and MEIS1-OE cells based on an LDA. MEIS1-OE cells had a significantly higher tumorigenic ability when inoculated into mice (Figure 4E), suggesting that MEIS1 is functionally important in the self-renewal activity of CSCs. In parallel, we monitored liver metastasis in a mouse model established by a splenic injection and found that MEIS1-OE cells had a greater ability than wild-type cells for metastatic outgrowth to the liver (Figure 4F,G). Since the mesenchymal phenotype is known to promote cancer stemness and metastasis [23], we conducted qPCR analyses to measure the expressions of mesenchymal and epithelial markers in MEIS1-OE cells. As a result, we confirmed the global increase in mesenchymal gene expression levels and decrease in epithelial gene expression levels upon MEIS1 overexpression (Appendix A). Our finding of a potential link between MEIS1 and the mesenchymal phenotype supports the conclusions in our manuscript that MEIS1 promotes cancer stemness (Figure 4E) and metastasis (Figure 4F,G). Collectively, our data provide the first evidence that MEIS1, a potential target gene of ICG-001, plays a critical role in CRC stemness and metastasis.

## 3. Discussion

This study first demonstrated the potent anti-CSC activity of ICG-001 against CRC in vitro and in vivo, providing preclinical evidence that ICG-001 exerts potent therapeutic effects on CRC metastasis. Moreover, through a series of bioinformatics and functional analyses, this study is the first to shed light on MEIS1, which promotes CSC properties and the malignancy of CRC, as a target gene of ICG-001.

Recently, many advances have been made in the development of small-molecule inhibitors that block the transcriptional activity of Wnt signaling [24]. Several previous studies have proven that ICG-001 inhibits the growth of CRC cells [11,25]. Beyond previous studies demonstrating the anti-growth effects of ICG-001 in CRC cells, this study strengthens the importance of ICG-001 by providing evidence that ICG-001 treatment efficiently attenuates CRC metastasis and CSC properties (Figure 2). Interestingly, we confirmed that the ICG-001-induced decrease in the sphere-forming ability persisted even when ICG-001 was removed from the culture (Figure 2C). This result supports a hypothetical mechanism by which this irreversible reduction in self-renewal activity might be linked to the reduction in tumor-initiating potential when ICG-001-treated HCT116 cells were reinoculated into mice without additional ICG-001 treatment (Figure 2D). In parallel, ICG-001 exhibited a potent inhibitory effect on liver metastasis in a mouse model established by a splenic injection by globally suppressing the expression of stemness- and metastasis-related genes (Figure 2E–G). Together, our findings suggest that ICG-001 is a potentially useful small molecule for the treatment of CRC metastasis, beyond its inhibitory effect on CRC growth.

Mechanistically, we found that MEIS1, a target gene of ICG-001, facilitated CRC malignancy by enhancing CSC properties (Figure 3 and Figure 4). MEIS1 is a developmentally conserved member of the 3-amino-acid loop extension family that can interact with homeobox proteins as a cofactor [26]. MEIS1 overexpression has been reported in leukemia [27] and neuroblastoma [28], suggesting its possible correlation with tumorigenesis. This study newly demonstrates the clinical implications of MEIS1 in CRC progression and recurrence (Figure 3), and highlights the functional importance of MEIS1 in CRC stemness and metastasis (Figure 4). Additionally, our findings suggest a hypothetical mechanism by which the elevation of MEIS1 expression in CSCs is mediated by the aberrant activation of Wnt signaling by demonstrating the significant reduction in MEIS1 expression induced by ICG-001 treatment (Figure 3B) and the knockdown of Wnt-signaling TFs (Figure 3C). Thus, additional studies to better understand how MEIS1 enhances CSC properties and how we can block the CSC-promoting function of MEIS1 would be valuable.

In summary, ICG-001 appears to robustly inhibit CRC metastasis and to reduce CSC properties. Although further studies are needed to reveal the precise mechanisms underlying its transcriptional and phenotypic activities, ICG-001 seems to broadly inhibit CRC stemness and metastasis by suppressing MEIS1 expression.

## 4. Materials and Methods

### 4.1. Bioinformatics Analysis

A Kaplan–Meier plot was generated with the R2 Platform (https://hgserver1.amc.nl/cgi-bin/r2/main.cgi, accessed on 15 September 2021), using the Sieber cohort (GSE14333) [29]. The DEG lists were obtained using Gene Expression Omnibus (GEO, GSE14333) by comparing Dukes’ stage A (*n* = 44) and D (*n* = 61) tumors or by comparing recurrent (*n* = 50) and nonrecurrent (*n* = 176) tumors. The GSEA was conducted as previously described [30]. A ranked GSEA was conducted with the DEG list using the Java implementation obtained from MSigDB (http://www.broadinstitue.org/gsea, accessed on 15 September 2021). The normalized enrichment score (NES) accounts for the differences in gene set sizes. The false discovery rate (FDR) q-value was used to set the significance threshold.

### 4.2. Cell Culture and Reagents

The HCT116, HT29, CCD-18Co, and SKOV3 cells were purchased from the Korean Cell Line Bank (Seoul, Republic of Korea). Cell culture and cell line authentication were performed as previously described [9]. ICG-001 was synthesized by Wuxi AppTec (Shanghai, China). The MEIS1 vector (EX-P0088-M68) and control empty vector (EX-NEG-M68) were purchased from GeneCopoeia (Rockville, MD, USA). The generation of luciferase-tagged HCT116 (HCT116-luc) cells was previously described [31].

### 4.3. Sphere Formation Assay

A sphere formation assay was performed as previously described [5]. The tumor sphere-forming efficiency (TSFE) was calculated as the number of spheres divided by the number of seeded cells. The detailed methods are provided in the online supplement.

### 4.4. Reverse Transcription–Quantitative Polymerase Chain Reaction (RT-qPCR)

The extraction of total RNA and RT-qPCR were performed as previously described [9]. The PCR was conducted using a StepOnePlus Real-Time PCR System (Applied Biosystems, Foster City, CA, USA). The list of PCR primer sequences is provided in the online supplement.

### 4.5. Western Blot Analysis

The protein isolation and Western blot analysis were conducted as previously described [9]. The detailed methods are provided in the online supplement.

### 4.6. Immunofluorescence Assay

The immunofluorescence assay was conducted as previously described [9]. The detailed methods are provided in the online supplement.

### 4.7. Animal Study

All animal experiments were conducted following approval from the Institutional Animal Care and Use Committee (IACUC) of Gwangju Institute of Science and Technology (GIST-2018-049). Male NOD.Cg-*Prkdc^scid^ Il2rg^tm1Wjl^*/SzJ (NSG) mice (Jackson Laboratory, Bar arbor, ME, USA) were used. To examine the self-renewal ability of CSCs in vivo, an LDA was conducted as described in our previous report [9]. To examine the metastatic potential, a mouse model was established by a splenic injection as described in our previous report [31]. The detailed methods are provided in the online supplement.

### 4.8. Small Interfering RNA (siRNA)-Mediated Knockdown

The siRNA-mediated knockdown was conducted as described in a previous report [9]. The list of siRNA sequences is provided in the online supplement. The efficacy tests of siRNAs for LEF1 and β-catenin were performed as described in our previous reports [8,30] and the most effective sequence was selected for further experiments. 

### 4.9. Statistical Analysis

All in vitro experiments were conducted in biological triplicate, and all in vivo experiments were conducted with five replicates per group. All statistical data are expressed as means ± standard deviations (SDs) for in vitro experiments and means ± standard errors of the mean (SEMs) for in vivo experiments. Statistical differences were determined using Student’s t-test for comparisons between two groups or one-way analysis of variance (ANOVA) with Dunnett’s multiple comparisons test for comparisons among three or more groups. The log-rank test was used for Kaplan–Meier analysis. The asterisks indicate statistical significance: *, ** and *** indicate *p* < 0.05, *p* < 0.01, and *p* < 0.001, respectively.

## Figures and Tables

**Figure 1 ijms-22-13413-f001:**
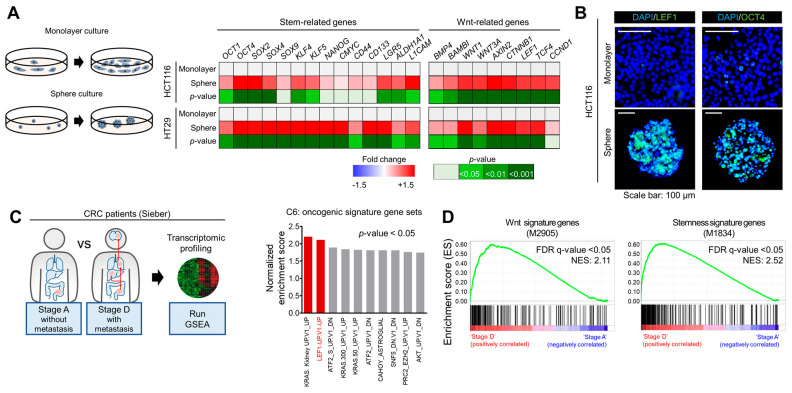
Relevance of Wnt activation and stemness to CRC metastasis. (**A**) RT−qPCR comparing the global trends in stemness- and Wnt-related gene expression between spheres and monolayer bulk cells. The data are presented as a heatmap with fold changes and *p*−values. (**B**) Immunofluorescence assay confirming the increased LEF1 and OCT4 protein levels in spheres. (**C**) Scheme for GSEA comparing Dukes’ stage A and D tumors from CRC patients. Top 10 oncogenic signature gene sets enriched in Dukes’ stage D CRC. (**D**) GSEA enrichment plots of Wnt signature genes (**left**) and stemness signature genes (**right**).

**Figure 2 ijms-22-13413-f002:**
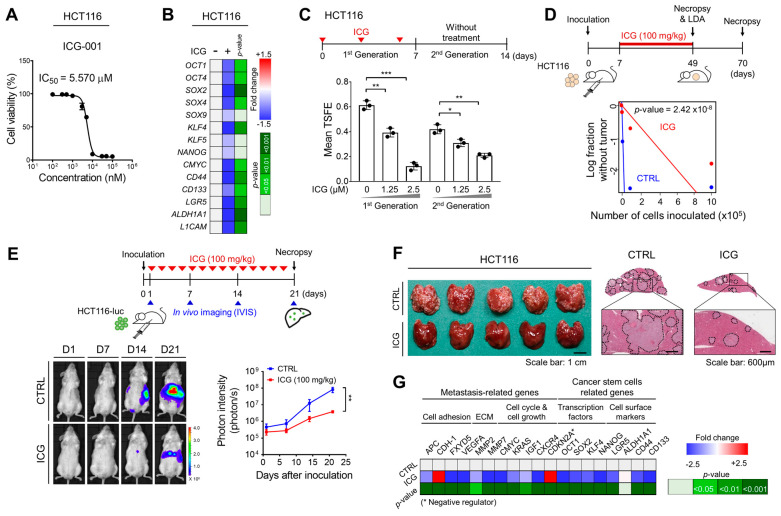
ICG−001, a Wnt inhibitor, disrupts CRC stemness and metastasis. (**A**) HCT116 cells were treated with increasing amounts of ICG−001 for 72 h, followed by MTT cell viability assays; (**B**) RT−qPCR confirming the global reductions in stemness-related gene expression levels by ICG−001 treatment (2.5 μM, 72 h). (**C**) Schematic view of the sphere formation assay measuring the effect of ICG−001 on the self-renewal activity of CSCs. A bar graph showing the reduction in the sphere-forming efficiency by ICG-001 treatment. (**D**) Schematic view (**top**) and results (**bottom**) of the LDA measuring the effect of ICG−001 on the self-renewal ability of CSCs. (**E**) Schematic view of the mouse model established by splenic injection to measure the effect of ICG−001 on CRC metastasis. The extent of liver metastasis was monitored by visualizing luciferase activity and (**F**) definitive necropsy. Representative images of gross anatomy and H&E-stained liver tissue. (**G**) RT−qPCR showing global reductions in metastasis- and CSC-related gene expression levels in ICG−001-treated metastatic colonies obtained from the livers of mice in the splenic injection model. *, ** and *** indicate *p* < 0.05, *p* < 0.01, and *p* < 0.001, respectively. CTRL, control; ICG, ICG−001.

**Figure 3 ijms-22-13413-f003:**
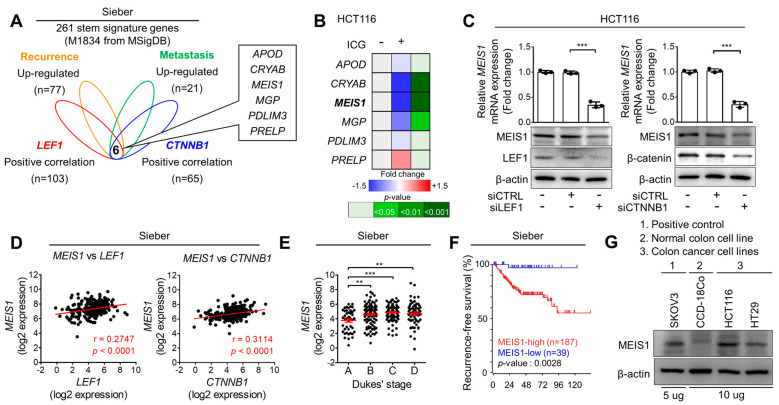
Identification and validation of MEIS1 as a potential target gene of ICG−001. (**A**) Identification of candidate target genes of ICG−001. Stemness signature genes were filtered by overlapping four gene sets obtained from the transcriptomic data of tumors from CRC patients. Six candidate genes were upregulated in both recurrent and metastatic tumors and positively correlated with LEF1 and β-catenin (CTNNB1) expression. (**B**) RT−qPCR validation performed after ICG−001 treatment (2.5 μM, 72 h). The data are presented as a heatmap with fold changes and *p*−values. (**C**) RT−qPCR and Western blot analyses conducted 48 h after LEF1 or β-catenin knockdown. (**D**) Gene expression correlations between MEIS1 and LEF1/β-catenin. (**E**) MEIS1 transcript levels according to the degree of CRC progression (Dukes’ stage). (**F**) Kaplan–Meier survival analyses of CRC patients based on MEIS1 expression. (**G**) Western blot analyses comparing the MEIS1 protein level in multiple human cell lines. ** and *** indicate *p* < 0.01 and *p* < 0.001, respectively.

**Figure 4 ijms-22-13413-f004:**
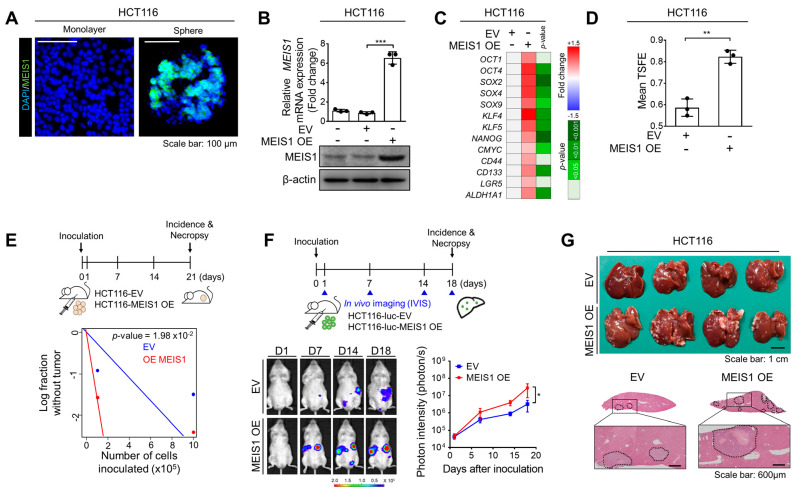
Functional validation of the effects of MEIS1 overexpression on CRC stemness and metastasis. (**A**) Immunofluorescence assay showing the increased MEIS1 protein level in spheres compared with bulk cells. (**B**) Validation of mRNA and protein levels in MEIS1-OE cells. (**C**) RT−qPCR confirming the global increases in stemness-related gene expression levels by MEIS1-OE. (**D**) Sphere formation assay comparing self-renewal ability between MEIS1-OE and control cells. (**E**) *In vivo* LDA comparing the effect of MEIS1-OE on the self-renewal ability of CSCs. (**F**,**G**) Mouse model established by splenic injection comparing metastatic potential between MEIS1-OE and control cells. The extent of liver metastasis was monitored by (**F**) visualizing luciferase activity and (**G**) definitive necropsy. Representative images of gross anatomy and H&E-stained liver tissue. ** and *** indicate *p* < 0.05, *p* < 0.01, and *p* < 0.001, respectively. CTRL, control; EV, empty vector; OE, overexpression.

## Data Availability

The datasets and methods used and/or analyzed in the current study are available within the manuscript or its supplementary information files. All data analyzed and materials used in this study are available from the corresponding author upon reasonable request.

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
