# Peer review of "The Small-Molecule Wnt Inhibitor ICG-001 Efficiently Inhibits Colorectal Cancer Stemness and Metastasis by Suppressing MEIS1 Expression"

_ijms, 2021, doi:10.3390/ijms222413413_

Round 1

Reviewer 1 Report

The manuscript entitled “The Small Molecule Wnt Inhibitor ICG-001 Efficiently Inhibits Colorectal Cancer Stemness and Metastasis by Suppressing MEIS1 Expression” sheds light on the importance of targeting the Wnt signaling to suppress CRC stemness and to reduce tumor growth and metastasis. Specifically, the authors provide evidences of the tumor suppressor function of ICG-001, a small inhibitor of the WNT signaling and of  its anti-metastatic potential. They also identified MEIS1 as a novel target gene of ICG-001 that promotes the CSC properties and metastasis in CRC and they confirmed these observations by in vitro and in vivo experiments.

The manuscript is well written and argued and all the experiments are detailed, but I think that some minor revisions are required:

  1. They authors demonstrated that the overexpression of MEF1 increases the stem properties and the tumorigenic potential of CRC cells. The authors should analyze if the downregulation of MEF1 (by using miRNA or shRNA constructs) counteracts the malignant behavior of tumor cells. For instance, they should perform a spheroid assay and check the number of the spheroids and size, that I suppose to be reduced in the downregulated spheres.
  2. Do the authors analyzed whether the MEF1 overexpression is associated with a more mesenchymal phenotype of the tumors cells? I thinks that they should analyze the expression of some markers (such as Vimentin, Snail1, Zeb1, N-cadherin and E-cadherin) by qPCR or western blotting.
  3. It is known that one of the main drivers of the stem properties of CRC cells is L1CAM (doi: 10.7150/thno.54027; doi: 10.1002/smll.202101711). The authors must analyze the expression of L1CAM in the spheres respect to adherent cells, to confirm that they are enriched in cancer stem cells. Moreover, they must also check if the treatment with ICG-001 reduces the expression of L1CAM and cite these papers.

Author Response

Please find the attached file "Answer to Reviewer comment_Reviewer1".

I attached the point-by-point response to your specific comments. 

Reviewer 2 Report

In this research, the authors demonstrated the efficacy of the small inhibitor of Wnt signaling ICG-001 in suppressing the tumorigenesis and metastatic abilities of human CRC cells.  Through a series of transcriptomic, bioinformatics, and in vivo mouse models for tumorigenesis/metastasis and CRC cell lines, they identified MEIS1 as the target gene of Wnt signaling that is affected by ICG-001.  The use of ICG-001 has proven to be effective in inhibiting Wnt signaling and stemness marker gene expression.  Further analysis of the mode of action of ICG-001 as a promising inhibitor of CRC development is worthwhile.

Author Response

Please find the attached file "Answer to Reviewer comment_Reviewer2".

I attached the point-by-point response to your specific comments. 
